# Global Existence of Bounded Solutions for Eyring–Powell Flow in a Semi-Infinite Rectangular Conduct

**Saeed ur Rahman [1], Jose Luis Diaz Palencia [2], Nomaq Tariq [1], Pablo Salgado Sánchez [3] and Julian Roa Gonzalez [2],\***

[1] Department of Mathematics, COMSATS University Islamabad, Abbottabad Campus, Abbottabad 22060, Pakistan
[2] Department of Education, Universidad a Distancia de Madrid, 28400 Madrid, Spain
[3] Spanish User Support and Operations Centre, Center for Computational Simulation, Universidad Politécnica de Madrid, 28223 Madrid, Spain
[*] Correspondence: julian.roa@udima.es

**Abstract:** The purpose of the present study is to obtain regularity results and existence topics regarding an Eyring–Powell fluid. The geometry under study is given by a semi-infinite conduct with a rectangular cross section of dimensions $L \times H$. Starting from the initial velocity profiles $(u_1^0, u_2^0)$ in $xy$-planes, the fluid flows along the $z$-axis subjected to a constant magnetic field and Dirichlet boundary conditions. The global existence is shown in different cases. First, the initial conditions are considered to be squared-integrable; this is the Lebesgue space $(u_1^0, u_2^0) \in L^2(\Omega)$, $\Omega = [0, L] \times [0, H] \times (0, \infty)$. Afterward, the results are extended for $(u_1^0, u_2^0) \in L^p(\Omega)$, $p > 2$. Lastly, the existence criteria are obtained when $(u_1^0, u_2^0) \in H^1(\Omega)$. A physical interpretation of the obtained bounds is provided, showing the rheological effects of shear thinning and shear thickening in Eyring–Powell fluids.

**Keywords:** nonlinear flow; Eyring–Powell fluid; geometrically three-dimensional flow; unsteady flow; global existence

## 1. Introduction

An Eyring–Powell fluid is a sub-class of a non-Newtonian fluid of interest in applied sciences. To cite some examples, we remark the applications in manufacturing engineering [1,2] and biological technology [3,4].

The Eyring–Powell model has been of interest for the description of magnetohydrodynamics (MHD). As representative of previous studies, Akbar et al. [5] carried out the analysis of solutions in a two-dimensional MHD fluid. Hina [6] considered an Eyring–Powell fluid for MHD purposes to study heat-transfer processes. Afterward, Bhatti et al. [7] proposed an analysis for a stretching surface under MHD physical principles. Similarly, other references can be cited describing analyses of Eyring–Powell fluids, combining analytical and numerical approaches, from purely mathematical principles to applications in different physical scenarios [8–19].

It shall be noted that there exists much literature dealing with the existence criteria of solutions when a fluid is formulated with the classical Newtonian viscosity involved in the Navier–Stokes equations; see the remarkable studies [20–29]. Nonetheless, the specific rheological properties of a fluid may lead to the exploration of other kinds of viscosity formulations. One of these formulations, based on the kinetic theory of liquids, led to the mentioned Eyring–Powell fluid. To the best of our knowledge, there is not a wide range of literature dealing with the existence and regularity of solutions in Eyring–Powell fluids in three-dimensional geometry. Consequently, our main objective is to introduce such an analysis under the most general conditions.

Considering some recent achievements related to the application of advanced analytical tools to non-Newtonian fluids, we can highlight the recent work of Bilal et al. [30], where the $(G'/G^2)-$expansion method was employed to obtain exact wave solutions to a Dullin–Gottwald–Holm system. In addition, the solutions to a Korteweg–deVries–Zakharov–Kuznetsov equation were explored in Ref. [31]. Based on a modified extended direct algebraic method, these authors found solutions in the form of solitary, shock, singular, shock singular, solitary shock and double singular solitons. In the present manuscript, we are concerned with the regularity and existence of solutions rather than with the specific forms of such solutions. This, however, establishes a basis for future research topics.

The paper layout is as follows. First, we introduce the framework of our study and describe the Eyring–Powell fluid model. Secondly, a set of three theorems is given so that their proofs permit to draw a conclusion on the regularity and existence of solutions to the proposed Eyring–Powell formulation. The introduced theorems are supported by a number of lemmas that are provided for the sake of clarity and by some propositions that are proved. The involved assessments follow a process that can be introduced sequentially as follows:

– Formulation of the involved equations in integral form.
– Derivation of a temporal differential equation in terms of spatial distributions in $L^p$ $(p \geq 2)$; the involved integrals are assessed, typically by parts.
– Introduction of hypotheses in a space of bounded mean oscillations that assure a bounded solution, and obtain the bounding constants.
– Application of the Gronwall theorem for a bound in the temporal law, and under spatial distributions in $L^p$ $(p \geq 2)$.

## 2. Model Formulation

We consider a flow of an electrically conducting Eyring–Powell fluid. The selection of this type of fluid is justified based on the following ideas. Firstly, the rheological properties of an Eyring–Powell fluid are derived based on the kinetic theory of liquids, instead of empirical or quasi-experimental principles; this can be the case of a Darcy–Forchheimer or a power–law fluid. Deducing a rheological law from a well-known theory makes the Eyring–Powell fluid interesting for purely mathematical assessments such that the analytical concepts rely on theoretical and well-proven physical aspects. Secondly, the Eyring–Powell rheological properties can be understood as an expansion of a typical linear fluid rheology. Then, the scope of our analysis contains some mathematical ideas that can be applied for the study of simpler rheological laws; this naturally extends to Newtonian fluids described by the classical continuity and momentum Navier–Stokes equations.

The Cartesian coordinates $(x, y, z)$, with the corresponding velocity components $V = (u_1, u_2, u_3)$, are chosen such that the origin is located in the plane sheet at $z = 0$. The fluid occupies the region $z > 0$, and flows from the sheet $z = 0$ to $z \rightarrow \infty$.

The conservation of mass and momentum are described in a general basis as

$$\nabla \cdot V = 0, \tag{1}$$

$$\rho \frac{dV}{dt} = \nabla \cdot \tau + J \times B, \tag{2}$$

where $dV/dt$ refers to the total derivative of the velocity field, $\rho$ is the fluid density, $B$ is the applied longitudinal (along the $z$-axis) magnetic field of magnitude $B_0$ driving the flow, $J$ is the current charges density, and $\tau$ refers to the Cauchy stress tensor, which is given by

$$\tau = -pI + \tau_{ij},$$

where $p$ is the pressure field in the fluid, $I$ is the identity tensor and $\tau_{ij}$ is the stress tensor typical in Eyring–Powell fluid models.

Based on the kinetic theory of liquids [32], a formulation to such stress tensor is

$$\tau_{ij} = \mu \frac{\partial u_i}{\partial x_j} + \frac{1}{\beta} \sinh^{-1}\left(\frac{1}{\gamma}\frac{\partial u_i}{\partial x_j}\right),$$

where $\mu$ is the dynamic viscosity, and $\beta$ and $\gamma$ are two characteristic constants related to the fluid spatial behavior and its characteristic relaxation frequency, respectively [33]. By considering

$$\sinh^{-1}\left(\frac{1}{\gamma}\frac{\partial u_i}{\partial x_j}\right) \approx \frac{1}{\gamma}\frac{\partial u_i}{\partial x_j} - \frac{1}{6}\left(\frac{1}{\gamma}\frac{\partial u_i}{\partial x_j}\right)^3 + \dots,$$

then

$$\tau_{ij} = \mu \frac{\partial u_i}{\partial x_j} + \frac{1}{\gamma\beta}\frac{\partial u_i}{\partial x_j} - \frac{1}{6\beta}\left(\frac{1}{\gamma}\frac{\partial u_i}{\partial x_j}\right)^3.$$

Note that we may consider higher-order terms, denoted by '...' in the expression above, when approximating the $\sinh^{-1}$ function, or even other forms of rheological behavior; see the work of Oke [34] for additional insights.

We assume a boundary layer is developed and analyze the velocity profiles in each $xy$-plane for which the following Dirichlet boundary conditions apply: $u_1 = u_2 = 0$ at $x = 0$, $L$ and $y = 0$, $H$.

Based on the exposed arguments and taking $L, \gamma^{-1}$ and $\rho$ as characteristic values for length, time and density, the governing equations written in dimensionless parameters read [32]

$$\frac{\partial u_1}{\partial t} + u_1 \frac{\partial u_1}{\partial x} + u_2 \frac{\partial u_1}{\partial y} + u_3 \frac{\partial u_1}{\partial z} =$$

$$\frac{1}{\mathrm{Re}}\left(1 + M - \frac{M}{2}\left(\frac{\partial u_1}{\partial z}\right)^2\right)\frac{\partial^2 u_1}{\partial z^2} - \mathrm{B}u_1, \tag{3}$$

$$\frac{\partial u_2}{\partial t} + u_1 \frac{\partial u_2}{\partial x} + u_2 \frac{\partial u_2}{\partial y} + u_3 \frac{\partial u_2}{\partial z} =$$

$$\frac{1}{\mathrm{Re}}\left(1 + M - \frac{M}{2}\left(\frac{\partial u_2}{\partial z}\right)^2\right)\frac{\partial^2 u_2}{\partial z^2} - \mathrm{B}u_2, \tag{4}$$

in the domain $\Omega = [0, 1] \times [0, \Gamma] \times (0, \infty)$, where $\Gamma = H/L$ refers to the cross-sectional aspect ratio, $\mathrm{Re} = \rho\gamma L^2/\mu$ is the Reynolds number, $\mathrm{M} = 1/(\gamma\beta\mu)$ characterizes the rheological behavior of the fluid, and $\mathrm{B} = \sigma B_0^2/(\rho\gamma) > 0$ is the dimensionless effective magnetic field inducing the flow. The kinematic Dirichlet boundary conditions now read $u_1 = u_2 = 0$ at $x = 0$, 1 and $y = 0$, $\Gamma$. Note that the own magnetic field generated by the charges motion is assumed to be negligible.

Further, the following conditions shall be considered as well:

$$\begin{aligned}
&\|u_1\|_{L^p([0,1]\times[0,\Gamma])}, \ \|u_2\|_{L^p([0,1]\times[0,\Gamma])} < \infty, \ \ \|u_3\|_{L^p([0,1]\times[0,\Gamma])} \ll \varepsilon_1 \ \text{at } z = 0,\\
&\|u_1\|_{L^p([0,1]\times[0,\Gamma])}, \ \|u_2\|_{L^p([0,1]\times[0,\Gamma])}, \ \|u_3\|_{L^p([0,1]\times[0,\Gamma])} \ll \varepsilon_2 \ \text{at } z \to \infty,\\
&u_1(x,y,z,0) = u_1^0, \ \ u_2(x,y,z,0) = u_2^0 \ \text{with} \ \ \|u_1^0\|_{L^p(\Omega)}, \|u_2^0\|_{L^p(\Omega)} < \infty,\\
&\|u_1^0\|_{L^p([0,1]\times[0,\Gamma])}, \ \|u_2^0\|_{L^p([0,1]\times[0,\Gamma])}, \ \ll \varepsilon_2 \ \text{at } z \to \infty,
\end{aligned} \tag{5}$$

where $p \geq 2$, $(u_1^0, u_2^0)$ are the initial velocity distributions in $xy$-planes and $0 < \varepsilon_2 \ll \varepsilon_1 \ll 1$.

## 3. Previous Definitions and Summary of Results

### 3.1. Previous Definitions and Results

Consider the Lebesgue norm $\|\cdot\|_{L^p}$ to define the functional space $L_p(\Omega)$. In addition, the usual Sobolev functional space of order $m$ is considered as

$$H^m(\Omega) = \left\{ u \in L^2(\Omega) : \nabla^m(u) \in L^2(\Omega) \right\},$$

with the norm

$$\|u\|_{H^m} = \left( \|u\|_{L^2}^2 + \|\nabla^m u\|_{L^2}^2 \right)^{\frac{1}{2}}.$$

As it will be specified later, we will establish the regularity criteria if $\|\partial u_1/\partial z\|_{\text{BMO}}^2$, $\|\partial u_2/\partial z\|_{\text{BMO}}^2$, $\|\nabla u_1\|_{\text{BMO}}^2$, and $\|\nabla u_2\|_{\text{BMO}}^2$ are sufficiently small. Note that BMO denotes the homogeneous space of 'bounded mean oscillations' associated with the norm

$$\|f\|_{\text{BMO}} \doteq \sup_{R^n, r > 0} \frac{1}{|B_r(x)|} \int_{B_r(x)} \left| f(y) - \frac{1}{|B_r(y)|} \int_{B_r(y)} f(z)dz \right| dy.$$

For further details on BMO spaces, we refer the reader to Ref. [35].

In addition, we recall the following two lemmas.

**Lemma 1.** *Let us consider $1 < q < p < \infty$, then*

$$\|u\|_{L^p} \leq A_1 \|u\|_{\text{BMO}}^{1-\frac{q}{p}} \|u\|_{L^q}^{\frac{q}{p}},$$

*where $A_1$ is a constant.*

For the proof of Lemma 1, we refer the reader to Ref. [35].

**Lemma 2.** *Given the functions $f$, $g$, $h \in C_c^\infty(\mathbb{R}^3)$, the following anisotropic inequality holds (see Ref. [22]):*

$$\left| \int_{\mathbb{R}^3} f\,g\,h\,dx\,dy\,dz \right| \leq A_2 \|f\|_q^{\frac{\alpha-1}{\alpha}} \|\partial_3 f\|_s^{\frac{1}{\alpha}} \|g\|_2^{\frac{\alpha-2}{\alpha}} \|\partial_1 g\|_2^{\frac{1}{\alpha}} \|\partial_2 g\|_2^{\frac{1}{\alpha}} \|h\|_2,$$

*where $A_2$ is a suitable constant, $\alpha > 2$, $1 \leq q, s \leq \infty$, and $(\alpha - 1)/q + 1/s = 1$.*

### 3.2. Statement of Results

The main results obtained in this analysis are stated as follows.

**Theorem 1.** *Assume that $(u_1^0, u_2^0) \in L^2(\Omega)$. In addition, assume that $\|\partial u_1/\partial z\|_{\text{BMO}}^2$ and $\|\partial u_2/\partial z\|_{\text{BMO}}^2$ are sufficiently small, then system (3)–(5) has a bounded global solution in $[0, T] \times \Omega$.*

**Theorem 2.** *Assuming that $(u_1^0, u_2^0) \in L^p(\Omega)$, then the system (3)–(5) has a global and bounded solution in $[0, T] \times \Omega$.*

**Theorem 3.** *Assume that $(u_1^0, u_2^0) \in H^1(\Omega)$, and that $\|\nabla u_1\|_{\text{BMO}}^2$, $\|\nabla u_2\|_{\text{BMO}}^2$, $\|\partial u_1/\partial z\|_{\text{BMO}}^2$, $\|\partial u_2/\partial z\|_{\text{BMO}}^2$ are sufficiently small, then the system (3)–(5) has a global and bounded solution in $[0, T] \times \Omega$.*

The proposed theorems are shown in the coming sections.

## 4. Proof of Theorem 1

The first intention is to show that the two-dimensional velocity profiles $(u_1, u_2)$ are globally bounded when the fluid is flowing through the $z$-axis. This means that for any

level in the *z*-axis, the fluid flow exhibits a regular behavior. The following proposition is required to support the proof of Theorem 1.

**Proposition 1.** *(Global bound of the two-dimensional velocity profile). Given the set of solutions $(u_1, u_2)$ to Equations (3)–(5) with initial distributions $(u_1^0, u_2^0)$ and in the assumption of $\|\partial u_1/\partial z\|_{BMO}^2, \|\partial u_2/\partial z\|_{BMO}^2 \ll 1$, the following global bound holds in $[0, T] \times \Omega$:*

$$\sup_{0 \leq t \leq T} \left( \|u_1\|_{L^2}^2 + \|u_2\|_{L^2}^2 \right) + C_3 \int_0^T \left[ \left\| \frac{\partial u_1}{\partial z} \right\|_{L^2}^2 + \left\| \frac{\partial u_2}{\partial z} \right\|_{L^2}^2 \right] dt$$

$$\leq C_4 \left( \left\| u_1^0 \right\|_{L^2}^2 + \left\| u_2^0 \right\|_{L^2}^2 \right), \qquad (6)$$

*where $C_3$ and $C_4$ are suitable constants related with the set of parameters involved in Equations (3)–(5).*

**Proof.** Multiplying Equation (3) by $u_1$ and integrating

$$\iiint_\Omega u_1 \frac{\partial u_1}{\partial t} dxdydz + I_1 = -\frac{1+M}{Re} \iiint_\Omega \left( \frac{\partial u_1}{\partial z} \right)^2 dxdydz$$

$$+ \frac{M}{6\,Re} \iiint_\Omega \left( \frac{\partial u_1}{\partial z} \right)^4 dxdydz - B \iiint_\Omega (u_1)^2 dxdydz,$$

which implies that

$$\frac{1}{2} \frac{d}{dt} \|u_1\|_{L^2}^2 + I_1 = -\frac{1+M}{Re} \left\| \frac{\partial u_1}{\partial z} \right\|_{L^2}^2 + \frac{M}{6\,Re} \left\| \frac{\partial u_1}{\partial z} \right\|_{L^2}^4 - B\|u_1\|_{L^2}^2, \qquad (7)$$

where

$$I_1 = \iiint_\Omega \left( u_1^2 \frac{\partial u_1}{\partial x} \right) dxdydz + \iiint_\Omega \left( u_1 u_2 \frac{\partial u_1}{\partial y} \right) dxdydz + \iiint_\Omega \left( u_1 u_3 \frac{\partial u_1}{\partial z} \right) dxdydz.$$

Note that we used Equation (1).

Developing further the integration on $I_1$:

$$I_1 = \iiint_\Omega u_1^2 \frac{\partial u_1}{\partial x} dxdydz - \iiint_\Omega \frac{u_1^2}{2} \frac{\partial u_2}{\partial y} dxdydz - \iiint_\Omega \frac{u_1^2}{2} \frac{\partial u_3}{\partial z} dxdydz$$

$$= -\iiint_\Omega \frac{u_1^2}{2} \left[ \frac{\partial u_2}{\partial y} + \frac{\partial u_3}{\partial z} \right] dxdydz$$

$$= \iiint_\Omega \frac{u_1^2}{2} \left( \frac{\partial u_1}{\partial x} \right) dxdydz = \int_0^\Gamma \int_0^\infty \left[ \frac{u_1^3}{6} \right]_0^1 dzdy.$$

Since $u_1 = 0$ at $x = 0, 1$, then $I_1 = 0$ and Equation (7) becomes

$$\frac{1}{2} \frac{d}{dt} \|u_1\|_{L^2}^2 = -\frac{1+M}{Re} \left\| \frac{\partial u_1}{\partial z} \right\|_{L^2}^2 + \frac{M}{6\,Re} \left\| \frac{\partial u_1}{\partial z} \right\|_{L^2}^4 - B\|u_1\|_{L^2}^2$$

$$\leq -\frac{1+M}{Re} \left\| \frac{\partial u_1}{\partial z} \right\|_{L^2}^2 + \frac{M\,C_1}{6\,Re} \left\| \frac{\partial u_1}{\partial z} \right\|_{L^2}^2 \left\| \frac{\partial u_1}{\partial z} \right\|_{BMO}^2 - B\|u_1\|_{L^2}^2,$$

where we used Lemma 1.

Now, provided that $\|\partial u_1/\partial z\|^2_{\text{BMO}}$ is sufficiently small, we can choose

$$C_1 \left\| \frac{\partial u_1}{\partial z} \right\|^2_{\text{BMO}} \leq C_2,$$

and therefore, the above equation becomes

$$\frac{1}{2}\frac{d}{dt}\|u_1\|^2_{L^2} \leq -\frac{1}{\text{Re}}\left(1 + M - \frac{M}{6}C_2\right)\left\|\frac{\partial u_1}{\partial z}\right\|^2_{L^2} - \text{B}\|u_1\|^2_{L^2},$$

which implies that

$$\frac{d}{dt}\|u_1\|^2_{L^2} + \frac{2}{\text{Re}}\left(1 + M - \frac{M}{6}C_2\right)\left\|\frac{\partial u_1}{\partial z}\right\|^2_{L^2} \leq -2\text{B}\|u_1\|^2_{L^2} \leq |\text{B}|\,\|u_1\|^2_{L^2}. \tag{8}$$

Similarly, multiplying Equation (4) by $u_2$ and integrating again, we obtain

$$\frac{d}{dt}\|u_2\|^2_{L^2} + \frac{2}{\text{Re}}\left(1 + M - \frac{M}{6}C_2\right)\left\|\frac{\partial u_2}{\partial z}\right\|^2_{L^2} \leq -2\text{B}\|u_2\|^2_{L^2} \leq |\text{B}|\,\|u_2\|^2_{L^2}. \tag{9}$$

Adding Equations (8) and (9):

$$\frac{d}{dt}\left(\|u_1\|^2_{L^2} + \|u_2\|^2_{L^2}\right) + \frac{2}{\text{Re}}\left(1 + M - \frac{M}{6}C_2\right)\left(\left\|\frac{\partial u_1}{\partial z}\right\|^2_{L^2} + \left\|\frac{\partial u_2}{\partial z}\right\|^2_{L^2}\right)$$
$$\leq \quad |\text{B}|\left(\|u_1\|^2_{L^2} + \|u_2\|^2_{L^2}\right).$$

In most realistic cases, the rheological parameter is small $|M| \ll 1$, and one can apply the Gronwall inequality to obtain

$$\sup_{0 \leq t \leq T}\left(\|u_1\|^2_{L^2} + \|u_2\|^2_{L^2}\right) + C_3 \int_0^T \left(\left\|\frac{\partial u_1}{\partial z}\right\|^2_{L^2} + \left\|\frac{\partial u_2}{\partial z}\right\|^2_{L^2}\right) dt \leq C_4\left(\left\|u_1^0\right\|^2_{L^2} + \left\|u_2^0\right\|^2_{L^2}\right),$$

in $[0, T] \times \Omega$, where $C_3$, $C_4$ depend on Re, and $M$ and $C_2$ should be upper bounded by $C_2 < 6(1 + M)/M$. $\quad\square$

From a physical point of view, this upper bound only applies to shear-thinning fluids with $M > 0$. This can be understood as a bound for the viscosity reduction that ensures an exponential or sub-exponential decrease in $(u_1, u_2)$ as $z \to \infty$ so that the associated integrals remain finite. For shear-thickening fluids with $M < 0$, in contrast, the increase in viscosity with the applied shear ensures such exponential (or sub-exponential) decay and, in practice, removes any condition on $C_2$. For additional insights about the rheological properties of Eyring–Powell fluids, the reader is referred to Ref. [34].

Note that the Theorem 1 is proved simply using the bound properties shown in Proposition 1.

### 5. Proof of Theorem 2

**Proof.** First, multiply Equation (3) by $|u_1|^{p-2} u_1$, $p > 2$, to make the following integration:

$$
\iiint_\Omega \frac{u_1^{p-1}}{p} \frac{\partial u_1^p}{\partial t} dxdydz + I_2 = -\frac{1+(p-1)M}{\mathrm{Re}} \iiint_\Omega \left( u_1^{\frac{p-2}{2}} \frac{\partial u_1}{\partial z} \right)^2 dxdydz
$$

$$
+ \frac{M}{6\mathrm{Re}} \iiint_\Omega \left( u_1^{\frac{p-2}{4}} \frac{\partial u_1}{\partial z} \right)^4 dxdydz - \mathrm{B} \iiint_\Omega u_1^p dxdydz
$$

$$
< -(p-1)\frac{1+M}{\mathrm{Re}} \iiint_\Omega \left( u_1^{\frac{p-2}{2}} \frac{\partial u_1}{\partial z} \right)^2 dxdydz
$$

$$
+ (p-1)\frac{M}{6\mathrm{Re}} \iiint_\Omega \left( u_1^{\frac{p-2}{2}} \frac{\partial u_1}{\partial z} \right)^4 dxdydz - \mathrm{B} \iiint_\Omega u_1^p dxdydz, \tag{10}
$$

where

$$
\begin{aligned}
I_2 &= \iiint_\Omega |u_1|^p \frac{\partial u_1}{\partial x} dxdydz + \iiint_\Omega |u_1|^{p-1} u_2 \frac{\partial u_1}{\partial y} + \iiint_\Omega |u_1|^{p-1} u_3 \frac{\partial u_1}{\partial z} dxdydz \\
&= -\iiint_\Omega \frac{u_1^p}{p} \frac{\partial u_2}{\partial y} dxdydz - \iiint_\Omega \frac{u_1^p}{p} \frac{\partial u_3}{\partial z} dxdydz \\
&= -\iiint_\Omega \frac{u_1^p}{p} \left( \frac{\partial u_2}{\partial y} + \frac{\partial u_3}{\partial y} \right) dxdydz \\
&= \iiint_\Omega \frac{u_1^p}{p} \left( \frac{\partial u_1}{\partial x} \right) dxdydz \\
&= \int_0^\Gamma \int_0^\infty \left[ \frac{u_1^{p+1}}{p(p+1)} \right]_0^1 dzdy.
\end{aligned}
$$

As $u_1 = 0$ at $x = 0, 1$, then $I_2 = 0$ and Equation (10) simplifies to

$$
\begin{aligned}
\frac{d}{dt}\|u_1\|_{L^p}^p &\leq -p(p-1)\frac{1+M}{\mathrm{Re}} \left\| u_1^{\frac{p-2}{2}} \frac{\partial u_1}{\partial z} \right\|_{L^2}^2 + p(p-1)\frac{M}{6\mathrm{Re}} \left\| u_1^{\frac{p-2}{2}} \frac{\partial u_1}{\partial z} \right\|_{L^4}^4 - p\mathrm{B}\|u_1\|_{L^p}^p \\
&\leq -p(p-1)\frac{1+M}{\mathrm{Re}} \left\| u_1^{\frac{p-2}{2}} \frac{\partial u}{\partial z} \right\|_{L^2}^2 \\
&\quad + C_5 p(p-1)\frac{M}{6\mathrm{Re}} \left\| u_1^{\frac{p-2}{2}} \frac{\partial u_1}{\partial z} \right\|_{L^2}^2 \left\| u_1^{\frac{p-2}{2}} \frac{\partial u_1}{\partial z} \right\|_{\mathrm{BMO}}^2 - p\mathrm{B}\|u_1\|_{L^p}^p.
\end{aligned}
$$

Since by initial assumption $\left\| u_1^{\frac{p-2}{2}} \partial u_1/\partial z \right\|_{\mathrm{BMO}}^2$ is sufficiently small, we can take $C_5 \left\| u_1^{\frac{p-2}{2}} \partial u_1/\partial z \right\|_{\mathrm{BMO}}^2 \leq C_6$, and choosing

$$
C_7 = \frac{p(p-1)}{\mathrm{Re}} \left( 1 + M - \frac{C_6 M}{6} \right),
$$

the following holds

$$
\frac{d}{dt}\|u_1\|_{L^p}^p + C_7 \left\| u_1^{\frac{p-2}{2}} \frac{\partial u_1}{\partial z} \right\|_{L^2}^2 \leq -p\mathrm{B}\|u_1\|_{L^p}^p \leq |\mathrm{B}|\|u_1\|_{L^p}^p.
$$

The application of the Gronwall inequality yields

$$\|u_1\|_{L^p}^p + C_8 \int_o^T \left\| u_1^{\frac{p-2}{2}} \frac{\partial u_1}{\partial z} \right\|_{L^2}^2 dt \leq C_9 \left\| u_1^0 \right\|_{L^p}^p,$$

where $C_8$ and $C_9$ refer to the Gronwall constants, compiling $C_6$ and $C_7$, that depend on the dimensionless parameters of the problem. Again, $C_6$ should obey the same upper bound derived for $C_2$.

Proceeding similarly, multiplying the Equation (4) by $|u_2|^{p-2}u_2, p > 2$, we obtain

$$\|u_2\|_{L^p}^p + C_8 \int_o^T \left\| u_2^{\frac{p-2}{2}} \frac{\partial u_2}{\partial z} \right\|_{L^2}^2 dt \leq C_9 \left\| u_2^0 \right\|_{L^p}^p,$$

with the same bounds for the involved constants. □

We recall the previous discussion about these bounds in the shear-thinning and shear-thickening cases.

## 6. Proof of Theorem 3

Before showing the Theorem 3, the following proposition is required.

**Proposition 2.** *Assume* $\left( \nabla u_1^0, \nabla u_2^0 \right) \in L^2(\Omega)$. *In addition, consider that* $\|\nabla u_1\|_{\mathrm{BMO}}^2, \|\nabla u_2\|_{\mathrm{BMO}}^2,$ $\|\partial u_1/\partial z\|_{\mathrm{BMO}}^2, \|\partial u_2/\partial z\|_{\mathrm{BMO}}^2$ *are sufficiently small, then the solution to the set of Equations (3)–(5) satisfies*

$$\left( \|\nabla u_1\|_{L^2}^2 + \|\nabla u_2\|_{L^2}^2 \right) + \frac{2}{\mathrm{Re}} \int_0^T \left( \left\| \frac{\partial}{\partial z} \nabla u_1 \right\|_{L^2}^2 + \left\| \frac{\partial}{\partial z} \nabla u_2 \right\|_{L^2}^2 \right) dt$$

$$+ \quad \frac{M}{2\mathrm{Re}} \int_0^T \left( \left\| \frac{\partial \nabla u_1}{\partial z} \frac{\partial u_1}{\partial z} \right\|_{L^2}^2 + \left\| \frac{\partial \nabla u_2}{\partial z} \frac{\partial u_2}{\partial z} \right\|_{L^2}^2 \right) dt \leq C_{14} \left[ \left\| \nabla u_1^0 \right\|_{L^2}^2 + \left\| \nabla u_2^0 \right\|_{L^2}^2 \right],$$

*where $C_{14}$ is a suitable constant related to the dimensionless parameters involved in the set of Equations (3)–(5) and the BMO bound hypothesis.*

**Proof.** Take the inner product in Equation (3) with $\Delta u_1$ and integrate with regards to the spatial variables to obtain

$$- \iiint_\Omega \nabla u_1 \frac{\partial \nabla u_1}{\partial t} (\nabla u_1)^2 dxdydz - \iiint_\Omega \nabla u_1 \left( \nabla \left( u_1 \frac{\partial u_1}{\partial x} + u_2 \frac{\partial u_1}{\partial y} + u_3 \frac{\partial u_1}{\partial z} \right) \right) dxdxdz$$

$$= \quad \frac{1+M}{\mathrm{Re}} \iiint_\Omega \frac{\partial}{\partial z} (\nabla u_1)^2 dxdydz - \frac{M}{2\mathrm{Re}} \iiint_\Omega \left( \frac{\partial u_1}{\partial z} \right)^2 \left( \frac{\partial \nabla u_1}{\partial z} \right)^2 dxdydz$$

$$+ \quad \mathrm{B} \iiint_\Omega (\nabla u_1)^2,$$

which implies that

$$- \frac{d}{dt} \|\nabla u_1\|_{L^2}^2 = I_3 - \frac{1+M}{\mathrm{Re}} \left\| \frac{\partial \nabla u_1}{\partial z} \right\|_{L^2}^2 - \frac{M}{2\mathrm{Re}} \left\| \frac{\partial u_1}{\partial z} \frac{\partial \nabla u_1}{\partial z} \right\|_{L^2}^2 - \mathrm{B} \|\nabla u_1\|_{L^2}^2,$$

where

$$I_3 \quad = \quad \iiint_\Omega \nabla \left( u_1 \frac{\partial u_1}{\partial x} + u_2 \frac{\partial u_1}{\partial y} + u_3 \frac{\partial u_1}{\partial z} \right) \nabla u_1 dxdydz$$

$$
\begin{aligned}
= \ & -\iiint_\Omega \left( u_1 \frac{\partial^2 u_1}{\partial x^2} + \left(\frac{\partial u_1}{\partial x}\right)^2 + u_2 \frac{\partial^2 u_1}{\partial x \partial y} + \frac{\partial u_1}{\partial y}\frac{\partial u_2}{\partial x} + u_3 \frac{\partial^2 u_1}{\partial x \partial z} + \frac{\partial u_1}{\partial z}\frac{\partial u_3}{\partial x} \right)\left(\frac{\partial u_1}{\partial x}\right) \\
& + \left( u_1 \frac{\partial^2 u_1}{\partial y \partial x} + \frac{\partial u_1}{\partial x}\frac{\partial u_1}{\partial y} + u_2 \frac{\partial^2 u_1}{\partial y^2} + \frac{\partial u_1}{\partial y}\frac{\partial u_2}{\partial y} + u_3 \frac{\partial^2 u_1}{\partial z \partial y} + \frac{\partial u_1}{\partial z}\frac{\partial u_3}{\partial y} \right)\left(\frac{\partial u_1}{\partial y}\right) \\
& + \left( u_1 \frac{\partial^2 u_1}{\partial z \partial x} + \frac{\partial u_1}{\partial x}\frac{\partial u_1}{\partial z} + u_2 \frac{\partial^2 u_1}{\partial y \partial z} + \frac{\partial u_1}{\partial y}\frac{\partial u_2}{\partial z} + u_3 \frac{\partial^2 u_1}{\partial z^2} + \frac{\partial u_1}{\partial z}\frac{\partial u_3}{\partial z} \right)\left(\frac{\partial u_1}{\partial z}\right) dx\,dy\,dz
\end{aligned}
$$

$$
\begin{aligned}
= \ & -\iiint_\Omega \left(\frac{\partial u_1}{\partial x}\right)^3 - \frac{\partial u_1}{\partial x}\frac{\partial u_1}{\partial y}\frac{\partial u_2}{\partial x} - \frac{\partial u_1}{\partial x}\frac{\partial u_1}{\partial z}\frac{\partial u_3}{\partial x} - \left(\frac{\partial u_1}{\partial y}\right)^2 \frac{\partial u_1}{\partial x} - \left(\frac{\partial u_1}{\partial y}\right)^2 \frac{\partial u_2}{\partial y} \\
& - \frac{\partial u_1}{\partial y}\frac{\partial u_1}{\partial z}\frac{\partial u_3}{\partial y} - \left(\frac{\partial u_1}{\partial z}\right)^2 \frac{\partial u_1}{\partial x} - \frac{\partial u_1}{\partial y}\frac{\partial u_1}{\partial z}\frac{\partial u_2}{\partial z} - \left(\frac{\partial u_1}{\partial z}\right)^2 \frac{\partial u_3}{\partial z} dx\,dy\,dz.
\end{aligned}
$$

Using Equation (1), we have

$$
\begin{aligned}
I_3 = \ & -\iiint_\Omega \left(\frac{\partial u_1}{\partial x}\right)^3 - \frac{\partial u_1}{\partial x}\frac{\partial u_1}{\partial y}\frac{\partial u_2}{\partial x} - \frac{\partial u_1}{\partial x}\frac{\partial u_1}{\partial z}\frac{\partial u_3}{\partial x} - \left(\frac{\partial u_1}{\partial y}\right)^2 \frac{\partial u_1}{\partial x} \\
& - \left(\frac{\partial u_1}{\partial y}\right)^2 \frac{\partial u_2}{\partial y} - \frac{\partial u_1}{\partial y}\frac{\partial u_1}{\partial z}\frac{\partial u_3}{\partial y} - \left(\frac{\partial u_1}{\partial z}\right)^2 \frac{\partial u_1}{\partial x} \\
& - \frac{\partial u_1}{\partial y}\frac{\partial u_1}{\partial z}\frac{\partial u_2}{\partial z} + \left(\frac{\partial u_1}{\partial z}\right)^2 \frac{\partial u_1}{\partial x} + \left(\frac{\partial u_1}{\partial z}\right)^2 \frac{\partial u_2}{\partial y} dx\,dy\,dz
\end{aligned}
$$

$$
\begin{aligned}
\leq \ & \iiint_\Omega \left|\frac{\partial u_1}{\partial x}\right|^3 + \left|\frac{\partial u_1}{\partial x}\right|\left|\frac{\partial u_1}{\partial y}\right|\left|\frac{\partial u_2}{\partial x}\right| + \left|\frac{\partial u_1}{\partial y}\right|^2 \left|\frac{\partial u_1}{\partial x}\right| + \left|\frac{\partial u_1}{\partial y}\right|^2 \left|\frac{\partial u_2}{\partial y}\right| + \left|\frac{\partial u_1}{\partial y}\right|\left|\frac{\partial u_1}{\partial z}\right|\left|\frac{\partial u_3}{\partial y}\right| \\
& + \left|\frac{\partial u_1}{\partial y}\right|\left|\frac{\partial u_1}{\partial z}\right|\left|\frac{\partial u_2}{\partial z}\right| + \left|\frac{\partial u_1}{\partial z}\right|^2 \left|\frac{\partial u_2}{\partial y}\right| dx\,dy\,dz
\end{aligned}
$$

$$
\begin{aligned}
\leq \ & \iiint_\Omega |\nabla u_1|^3 + |\nabla u_1|^2 |\nabla u_2| + |\nabla u_1|^3 + |\nabla u_1|^2 |\nabla u_2| + |\nabla u_1|^2 |\nabla u_3| + |\nabla u_1|^2 |\nabla u_2| \\
& + |\nabla u_1|^2 |\nabla u_2| dx\,dy\,dz \\
= \ & \iiint_\Omega \left( 2|\nabla u_1|^3 + 4|\nabla u_1|^2 |\nabla u_2| + |\nabla u_1|^2 |\nabla u_3| \right) dx\,dy\,dz,
\end{aligned}
$$

where $|\partial u_i / \partial x_j| \leq |\nabla u_i|$.

Since $|\nabla u_3|$ is very small, we can choose $|\nabla u_3| \leq C_{10}$ to obtain

$$
I_3 \leq \iiint_\Omega \left( 2|\nabla u_1|^3 + 4|\nabla u_1|^2 |\nabla u_2| + C_{10}|\nabla u_1|^2 \right) dx\,dy\,dz
$$

$$
= \iiint_\Omega \left( 2|\nabla u_1|^2 |\nabla u_1| + 4|\nabla u_1|^2 |\nabla u_2| + C_{10}|\nabla u_1|^2 \right) dx\,dy\,dz.
$$

Considering now the Young's inequality

$$
\begin{aligned}
2 & \iiint_\Omega \left[ \frac{1}{2}(\nabla u_1)^4 + \frac{1}{2}(\nabla u_1)^2 \right] dxdydz + 4 \iiint_\Omega \left[ \frac{1}{2}(\nabla u_1)^4 + \frac{1}{2}(\nabla u_2)^2 \right] dxdydz \\
+ & \; C_{10} \iiint_\Omega (\nabla u_1)^2 dxdydz \\[4pt]
= & \iiint_\Omega 2(\nabla u_1)^4 dxdydz + \iiint_\Omega (\nabla u_1)^2 dxdydz + \iiint_\Omega (\nabla u_2)^2 dxdydz \\
+ & \; C_{10} \int (\nabla u_1)^2 dxdydz \\[4pt]
= & \; 2\|\nabla u_1\|_{L^4}^4 + (1 + C_{10})\|\nabla u_1\|_{L^2}^2 + \|\nabla u_2\|_{L^2}^2 \\[4pt]
\leq & \; 2\|\nabla u_1\|_{L^2}^2 \|\nabla u_1\|_{BMO}^2 + (1 + C_{10})\|\nabla u_1\|_{L^2}^2 + \|\nabla u_2\|_{L^2}^2,
\end{aligned}
$$

where we used Lemma 1.

Since $\|\nabla u_1\|_{\text{BMO}}$ is sufficiently small for our purposes, we can choose $\|\nabla u_1\|_{\text{BMO}} \leq C_{11}$. Therefore, $I_3$ becomes

$$
\begin{aligned}
I_3 & \leq (1 + C_{10} + 2C_{11})\|\nabla u_1\|_{L^2}^2 + \|\nabla u_2\|_{L^2}^2 \\[6pt]
& \leq C_{12}\left( \|\nabla u_1\|_{L^2}^2 + \|\nabla u_2\|_{L^2}^2 \right),
\end{aligned}
$$

where $C_{12} = (1 + C_{10} + 2C_{11})$.

Then, we arrive to

$$
\begin{aligned}
\frac{d}{dt}\|\nabla u_1\|_{L^2}^2 & + \frac{1 + M}{\text{Re}}\left\| \frac{\partial \nabla u_1}{\partial z} \right\|_{L^2}^2 + \frac{M}{2\text{Re}}\left\| \frac{\partial u_1}{\partial z}\frac{\partial \nabla u_1}{\partial z} \right\|_{L^2}^2 \\[6pt]
& \leq (C_{12} - B)\|\nabla u_1\|_{L^2}^2 + \|\nabla u_2\|_{L^2}^2 \\
& \leq C_{13}\left( \|\nabla u_1\|_{L^2}^2 + \|\nabla u_2\|_{L^2}^2 \right),
\end{aligned}
\tag{11}
$$

where $C_{13} = C_{12} - B > 0$.

Similarly, multiplying Equation (4) by $\nabla u_2$ and after integration by parts, we have

$$
\begin{aligned}
\frac{d}{dt}\|\nabla u_2\|_{L^2}^2 & + \frac{1 + M}{\text{Re}}\left\| \frac{\partial \nabla u_2}{\partial z} \right\|_{L^2}^2 + \frac{M}{2\,\text{Re}}\left\| \frac{\partial u_2}{\partial z}\frac{\partial \nabla u_2}{\partial z} \right\|_{L^2}^2 \\[6pt]
& \leq C_{13}\left( \|\nabla u_1\|_{L^2}^2 + \|\nabla u_2\|_{L^2}^2 \right).
\end{aligned}
\tag{12}
$$

Adding Equations (11) and (12),

$$
\begin{aligned}
\frac{d}{dt}&\left[ \|\nabla u_1\|_{L^2}^2 + \|\nabla u_2\|_{L^2}^2 \right] + \frac{1 + M}{\text{Re}}\left[ \left\| \frac{\partial}{\partial z}\nabla u_1 \right\|_{L^2}^2 + \left\| \frac{\partial}{\partial z}\nabla u_2 \right\|_{L^2}^2 \right] \\[6pt]
+ & \; \frac{M}{2\,\text{Re}}\left[ \left\| \frac{\partial \nabla u_1}{\partial z}\frac{\partial u_1}{\partial z} \right\|_{L^2}^2 + \left\| \frac{\partial \nabla u_2}{\partial z}\frac{\partial u_2}{\partial z} \right\|_{L^2}^2 \right] \leq C_{13}\left( \|\nabla u_1\|_{L^2}^2 + \|\nabla u_2\|_{L^2}^2 \right).
\end{aligned}
$$



Finally, the Gronwall inequality yields

$$
\left( \|\nabla u\|_{L^2}^2 + \|\nabla u\|_{L^2}^2 \right) + \frac{2}{\mathrm{Re}} \int_0^T \left( \left\| \frac{\partial}{\partial z} \nabla u_1 \right\|_{L^2}^2 + \left\| \frac{\partial}{\partial z} \nabla u_2 \right\|_{L^2}^2 \right) dt
$$

$$
+ \quad \frac{2\,M}{\mathrm{Re}} \int_0^T \left( \left\| \frac{\partial \nabla u_1}{\partial z} \frac{\partial u_1}{\partial z} \right\|_{L^2}^2 + \left\| \frac{\partial \nabla u_2}{\partial z} \frac{\partial u_2}{\partial z} \right\|_{L^2}^2 \right) dt \le C_{14} \left[ \left\| \nabla u_1^0 \right\|_{L^2}^2 + \left\| \nabla u_2^0 \right\|_{L^2}^2 \right],
$$

where $C_{14}$ depends on the dimensionless parameters of the problem. □

Theorem 3 is shown by making use of the results obtained in Propositions 1 and 2.
Compared to the previous results, the unique bound required here is $C_{12} > B$, which can be understood as a bound for the applied magnetic field.

## 7. Conclusions

In this paper, we developed the global existence of regular solutions for an Eyring–Powell fluid flowing along a semi-infinite conduct with a rectangular cross-section of dimensions $[0, 1] \times [0, \Gamma]$, subjected to a constant longitudinal magnetic field of (dimensionless) magnitude B. The initial velocity profiles $(u_1^0, u_2^0)$ were given in $xy$-planes along the $z$-axis, and the flow developed in the region $z > 0$. The following results were provided. Firstly, for $(u_1^0, u_2^0) \in L^2(\Omega)$, $\Omega = [0, 1] \times [0, \ \Gamma] \times (0, \infty)$, a regular global solution was shown to hold. A similar existence result was proved in the case $(u_1^0, u_2^0) \in L^p(\Omega)$, $p > 2$. Finally, we obtained similar existence criteria for $(u_1^0, u_2^0) \in H^1(\Omega)$.

The proposed results can be of practical use to support the resolution of the Eyring–Powell fluid with numerical means. Prior to starting any numerical assessment, the regularity of the solutions can be interpreted based on the results outlined in this work. As a future research topic related to the proposed Eyring–Powell fluid, one can consider the possibility of understanding the behavior of the solutions together with their increasing or decreasing rate. A remarkable question to explore is related to the existence of an exponential profile for a special class of solutions known as traveling waves. The fact of having an exponential behavior leads to state the regularity of the solutions, and shall be compliant with the obtained results.

**Author Contributions:** Conceptualization, S.u.R., J.L.D.P., N.T., P.S.S. and J.R.G.; methodology, S.u.R., J.L.D.P., N.T., P.S.S. and J.R.G.; validation, S.u.R., J.L.D.P., N.T., P.S.S. and J.R.G.; formal analysis, S.u.R., J.L.D.P., N.T., P.S.S. and J.R.G.; investigation, S.u.R., J.L.D.P., N.T., P.S.S. and J.R.G.; resources, S.u.R., J.L.D.P., N.T., P.S.S. and J.R.G.; writing—original draft preparation, S.u.R., J.L.D.P., N.T., P.S.S. and J.R.G.; writing—review and editing, S.u.R., J.L.D.P., N.T., P.S.S. and J.R.G.; funding acquisition, J.L.D.P. and J.R.G. All authors have read and agreed to the published version of the manuscript.

**Funding:** This research received no external funding.

**Institutional Review Board Statement:** Not applicable.

**Informed Consent Statement:** Not applicable.

**Data Availability Statement:** Not applicable.

**Conflicts of Interest:** The authors declare no conflict of interest.

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
