# Peer review of "Global Existence of Bounded Solutions for Eyring–Powell Flow in a Semi-Infinite Rectangular Conduct"

_axioms, doi:10.3390/axioms11110625_

Round 1
Reviewer 1 Report
This manuscript is very interesting. It is written on the hot topic of non-Newtonian fluid hydrodynamics. The paper presents regularity and existence theorems for solutions describing the Eyring-Powell fluid flow for the boundary layer equation. The article can be published because it is of interest to the reader, but the authors need to make changes to the text of the manuscript.
1. Why was the Eyring-Powell fluid chosen for studying the properties of solutions of non-Newtonian fluids?
2. Why were only two terms taken into account when approximating the rheological law?
3. Is the proof fair if only one term in the rheological law is taken into account?
4. In the introduction, it is necessary to give a bibliographic review not only on theorems, but also to take into account the exact solutions of the Navier-Stokes equations.
Author Response
Dear reviewer, thanks you for your comments. Please view the file attached.

Reviewer 2 Report
Please find below my suggested comments:
1- The authors are expected to explain more about Eyring-Powel fluids and discuss the existing solutions in the literature showing the research gap that the current work is exploring it.
2- Although the authors tried to show the framework of the research and how it is organized within lines 35-40, it is recommended to demonstrate the framework of the study in a schematic way or with a flow chart to be more comprehensible.
3- Authors are recommended to verify the proposed solutions with an existing experimental example and data.
4- Authors are expected to elaborate more in the conclusion part (do not use the terms same or similar results without giving sufficient content about your statement). What are the potential physical applications are referring to (please explain with an example)?
Author Response

(The authors gave the same response as above.)

Reviewer 3 Report
Derived results are correct and interesting. However, in order to publish in a reputed journal, authors need to incorporate the following suggestions to improve the quality of the manuscript.
1. Abstract should be rewritten and extended so that it can reflect the overall contain of the paper.
2. Check the manuscript carefully for typos and grammatical errors.
3. Future research direction must be shown in conclusion.
4. References are very less in number please cite the paper given in 7.
5. Explain your original contribution.
6. Compare your results with existing methods. You have compared results by three polynomials.
7. References should be written in unique way.
8. The authors should give recent development in the area (operational matrix method) and add the following references:
Mathematical Methods in the Applied Sciences 44, (5), (2021) 4094-4104.
Author Response

(The authors gave the same response as above.)

Round 2
Reviewer 3 Report
Accepted in new version